# A Recursive Prediction-Based Feature Enhancement for Small Object Detection

**DOI:** 10.3390/s24123856

**Published:** 2024-06-14

**Authors:** Xiang Xiao, Xiaorong Xue, Zhiyuan Zhao, Yisheng Fan

**Affiliations:** School of Electronics and Information Engineering, Liaoning University of Technology, Jinzhou 121001, China; 219807021@stu.lnut.edu.cn (X.X.); 219807032@stu.lnut.edu.cn (Z.Z.); 219915003@stu.lnut.edu.cn (Y.F.)

**Keywords:** small object detection, SAC, DINO, NWD

## Abstract

Transformer-based methodologies in object detection have recently piqued considerable interest and have produced impressive results. DETR, an end-to-end object detection framework, ingeniously integrates the Transformer architecture, traditionally used in NLP, into computer vision for sequence-to-sequence prediction. Its enhanced variant, DINO, featuring improved denoising anchor boxes, has showcased remarkable performance on the COCO val2017 dataset. However, it often encounters challenges when applied to scenarios involving small object detection. Thus, we propose an innovative method for feature enhancement tailored to recursive prediction tasks, with a particular emphasis on augmenting small object detection performance. It primarily involves three enhancements: refining the backbone to favor feature maps that are more sensitive to small targets, incrementally augmenting the number of queries for small objects, and advancing the loss function for better performance. Specifically, The study incorporated the Switchable Atrous Convolution (SAC) mechanism, which features adaptable dilated convolutions, to increment the receptive field and thus elevate the innate feature extraction capabilities of the primary network concerning diminutive objects. Subsequently, a Recursive Small Object Prediction (RSP) module was designed to enhance the feature extraction of the prediction head for more precise network operations. Finally, the loss function was augmented with the Normalized Wasserstein Distance (NWD) metric, tailoring the loss function to suit small object detection better. The efficacy of the proposed model is empirically confirmed via testing on the VISDRONE2019 dataset. The comprehensive array of experiments indicates that our proposed model outperforms the extant DINO model in terms of average precision (AP) small object detection.

## 1. Introduction

Object detection technologies, underpinned by Convolutional Neural Network (CNN), have seen significant advancements in recent years and are increasingly deployed for object detection across a variety of scenarios. Established paradigms of object detection such as Faster R−CNN, Single Shot Multibox Detector (SSD), and You Only Look Once (YOLO) [1,2,3], utilize deep neural networks at their core to extract image features. They employ region proposal methods to identify potential object areas for subsequent classification and localization. These potential regions require object classification and position regression to determine the final object bounding boxes and categories. Unlike conventional object detection methods that rely on region proposals [4], Detection Transformer (DETR) [5] is an innovative detection algorithm based on the Transformer architecture. It eliminates the need for manually designed components and achieves performance comparable to optimized traditional detectors like Faster RCNN. DETR combines Transformer with CNN−based [6,7,8] backbone models for feature extraction, employing an encoder–decoder architecture derived from the transformer [9] framework. This setup accurately classifies and precisely locates object positions within images. In addition, DETR models object detection as a set prediction task and utilizes bipartite graph matching for label assignment. It employs learnable queries to detect the presence of objects and integrates features within the image feature map.

Many improved methodologies akin to DETR have been proposed to accelerate training convergence and enhance the effectiveness of queries in DETR [10,11,12]. Particularly, DeNoising Anchor Box (DN−DETR) [13] further resolves the instability of bipartite matching by introducing denoising (DN) technology and the effectiveness of this approach has been proven by results on the COCO 2017 dataset, affirming that transformer-based object detection models are capable of achieving outstanding performance.

Detecting small objects with deep neural networks is challenging because they represent a minimal pixel area in images, leading to reduced feature extraction by the network [14]. The key issue in small object detection is utilizing the limited feature information to precisely extract semantic and positional details of small−scale targets. While methods building on Faster R−CNN, YOLOv5, and FCOS [15] have improved small object detection, they often neglect contextual information and complicate network design. DETR detectors, particularly for small object detection, have seen less development in this area. This paper introduces an approach that enhances the effectiveness of small object detection by incorporating ideas from Deformable DETR, SQR DETR [16], and DINO [17], refining the backbone architecture and loss functions and introducing a recursive feature enhancement method.

Many top−performing transformer−based models use ResNet50 as the backbone network to alleviate the vanishing and exploding gradient issue in deep networks through residual connections. However, ResNet50 [6] does not specifically cater to small objects, which leads to inferior performance during the detection of small−sized targets. Switchable Atrous Convolution [18] (SAC), a variant of atrous convolution, adopts a skipped sampling method that expands the convolutional kernel’s field of view. Unlike traditional convolution operations, dilated convolution [19] has fixed gaps, or dilation rates, between kernel elements. By altering the dilation rate, the receptive field [20] is enlarged without decreasing the resolution, effectively enhancing detection and classification performance for small objects.

The loss function quantifies the model’s prediction errors against true labels, serving as an optimizable objective for parameter tuning. In object detection, classification loss assesses bounding box classification accuracy, while regression loss measures the location discrepancy of these boxes. In the DETR model, the design of the loss function is a linear combination of classification and coordinate regression losses, without specifically targeting improvements for small objects. GIOU loss [21], which is used in coordinate regression, has a clear flaw: it is much more sensitive to object scale variance. To enhance model performance to small object detection, a new metric, Normalized Wasserstein Distance (NWD) [22], has been introduced into the overall coordinate loss to measure the similarity between bounding boxes, and it is mixed with GIOU loss for calculations.

In CNN−based detectors, the Feature Pyramid Network (FPN) [23] is a critical component that integrates multi−layer features to improve image resolution and detail sensitivity. However, considerable computational costs and memory footprints have rendered FPN less favored in transformer-based models. In light of this, we have devised a recursive prediction approach to bolster the detection of small target areas. This method is applied to the output of each transformer decoder layer and utilizes a secondary filtering process to select features characteristic of small objects. Research has indicated that subsequent layers of the transformer decoder exhibit a reduced capacity to perceive finer details and smaller objects, resulting in these items not being effectively detected. Therefore, before conducting Hungarian matching, our model recursively feeds the results of each transformer decoder layer back into the backbone network for feature enhancement.

The main contributions of this paper are as follows:In our study, we scrutinize DETR’s backbone network aimed at feature extraction, opting to enhance it with a technique called Switchable Atrous Convolution, supplanting the standard 3 × 3 convolutions. This versatile method dynamically adjusts the receptive field and assimilates global context, significantly boosting the ability to detect small objects.To mitigate the deficiencies of DETR’s decoder in handling small object detection, we introduce a novel Recursive Small Object Prediction (RSP) module. For implementation, the RSP module initially filters decoder predictions based on classification scores and object area, mapping the sifted tokens back onto the backbone’s low−dimensional feature layers. It then refines these initial results to enable more exhaustive small object feature extraction, resulting in an augmented tally of small object predictions.Due to DETR’s lack of consideration for small objects in its loss function, we integrate the Normalized Wasserstein Distance (NWD) metric to refine the loss calculation, and through experimentation, we confirm its effectiveness. NWD loss presents a refined metric for gauging similarity, particularly amongst small objects, enhancing the precision of detection.

## 2. Related Work

### 2.1. CNN−Based Object Detection

Due to their formidable feature extraction capabilities, Convolutional Neural Networks (CNNs) have become a staple in the field of computer vision. They are broadly bifurcated into two paradigms: two−stage and one−stage object detection. Algorithms like Faster RCNN [1] represent the two−stage approach, dividing the object detection process into region proposal and object classification with localization. The region proposal stage generates potential target−containing candidate regions using methods such as Selective Search or Region Proposal Networks (RPN). The features from these candidate boxes are then retrieved via Region of Interest (ROI) pooling operations for subsequent classification and regression tasks. On the other hand, one−stage object detection models like YOLO [3] streamline the process by discarding proposal generation and making direct predictions on the entire image. Utilizing predefined anchors, these models localize targets directly and output precise positional and categorical information of the objects.

Current convolution−based approaches to object detection have reached a level of maturity and widespread application. Yet, the efficacy of these models often hinges on manually designed components like anchor boxes, which demand prior knowledge specific to the detection task, and the application of Non−Maximum Suppression (NMS) [24] to refine the target boxes. In contrast, the DETR model leverages self−attention mechanisms within a transformer network to facilitate end−to−end object detection. Its innovatively designed loss function, referred to as the Hungarian matching loss, adeptly resolves challenges associated with NMS. It achieves this by optimally pairing predicted boxes with ground−truth boxes through bipartite graph matching, thereby streamlining the matching process. By directly refining this process, DETR adeptly handles overlapping objects, rendering traditional NMS steps unnecessary. This groundbreaking loss function design propels DETR to notable success in object detection tasks, showcasing its exceptional performance.

### 2.2. Small Object Detection

Similar to other object detection models, methods for small/tiny object detection can be divided into three core categories [25,26,27,28]: multi−scale feature learning, designing better training approaches, and GAN−based detection.

Multi−scale feature learning is a straightforward yet traditional approach that involves resizing input images to various scales and training distinct detectors optimized for specific scale ranges. To reduce the computational costs, feature pyramids at different scales have been developed. For instance, SSD [2] detects objects from feature maps of varying resolutions. The Feature Pyramid Network (FPN) constructs a top−down architecture with lateral connections, merging feature information across scales to enhance object detection performance.

Designing better training strategies can significantly enhance the detection of small objects in computer vision tasks. DetectoRS [18] features a Macro Design with a Recursive Feature Pyramid (RFP) that introduces feedback connections from the top−down FPN layers to the bottom−up backbone layers. Meanwhile, CZDet [29] and Sparse R−CNN [30] are exemplars of exceptional training strategies.

Object detection using Generative Adversarial Network (GAN) presents a novel method where two neural networks, the generator and the discriminator, engage in adversarial training. The generator aims to produce images resembling real ones, whereas the discriminator distinguishes between real and generated images. GAN-based models like Perceptual GAN [31] and MT-GAN [32] enhance small object detection; the former by reducing differences between small and large objects, and the latter by training a super-resolution model to improve small RoIs features.

Transformer are becoming increasingly influential in object detection, with models like Vision Transformer (ViT) [33] and Swin Transformer [34] setting benchmarks. Variants of DETR, such as Deformable DETR, use multi−scale deformable attention to surpass the original DETR performance. DAB−DETR [35] explores decoder improvements by introducing dynamic anchor boxes for refined cross−attention computation. DN−DETR [13] presents a denoising strategy to stabilize bipartite graph matching, speeding up model convergence. DINO enhances DETR variants with a contrastive denoising training method, mixed query selection for anchoring, and forward−twice bounding box prediction. Our RSP enhances DINO’s detection of small objects through a two−stage filtering improvement.

## 3. Methods

Figure 1 illustrates our proposed architecture, which ingeniously integrates Switchable Atrous Convolution to enrich understanding of the context and background surrounding small objects. Additionally, it incorporates the NWD metric, augmenting geometric perception of small objects. Our designed RSP component is utilized to bolster the final predictions for small targets by filtering preliminary decoder outputs, subsequently channeling relevant token data back into the backbone and then flattening this information to be guided into the encoder.

### 3.1. Switchable Atrous Convolution

In traditional deep neural networks, increasing the number of layers often leads to performance saturation or even degradation, which means that adding more layers does not always result in improved outcomes. ResNet addresses this issue by introducing residual blocks. Each residual block enables the network to learn a residual function, representing the difference between the target mapping and the input. Although ResNet is efficaciously implemented across various computer vision tasks, the performance in detecting small objects is often impeded by the inherently small receptive fields of traditional convolutional layers, which may not capture fine details of smaller objects adequately. To surmount this challenge, we integrate SAC to expand the receptive field. Atrous convolution allows for increased receptive field size and preserves feature resolution by adjusting the dilation rate. This enhancement improves the network’s ability to detect and classify small objects while also providing richer context and background information. This is achieved without an uptick in the number of network parameters or computational complexity, which leads to improved detection of small objects within the constraints of the existing network architecture.

Figure 2 illustrates the SAC structure and how global context modules are incorporated before and after SAC to integrate image−level information into the features, highlighting its three main components. Specifically, the core concept of SAC involves convolving the same input features with different dilation rates and fusing the outcomes of these varied convolutions through a specially designed switch function. Assume a conventional convolutional layer, denoted as *y* = Conv(*x*, *w*, *r*), where *w* represents the weights, *r* represents the dilation rate, *x* is the input, and *y* is the output. This convolutional layer can be transformed into SAC according to the following formula:(1)$$Conv(x,w,1)\overset{Convert}{\underset{to\;SAC}{\to }}S(x)\cdot Conv(x,w,1)+(1-S(x))\cdot Conv(x,w+\Delta w,r)$$
where Δ*w* represents a learnable weight, wiht the transformation function *S*(·) consists of a 5 *×* 5 average pooling layer followed by a 1 *×* 1 convolutional layer, which is relative to both the input and its position.

To address the issue of weight initialization, a locking mechanism is proposed. Specifically, we set one weight to w and another to *w* + Δ*w*. The rationale behind this is that object detection commonly employs pretrained models for initialization. However, for SAC, which is derived from standard convolutional layers, the weights with larger atrous rates are typically missing. Since different atrous rates can be employed under the same weight to coarsely detect targets of varying sizes, the weights of pretrained models may be utilized to initialize the missing weights. There are two lightweight global context modules before and after SAC; this is attributable to the input features having been processed by a global average pooling layer. The weights of these two modules are minimal. These global context modules resemble the Squeezeand−Excitation Network (SENet) [36] but differ in two key aspects: the global context modules comprise solely a convolutional layer without any non−linear layers, and the output is added back to the backbone network instead of being multiplied with the input.

### 3.2. Recursive Small Object Prediction

Within the DETR series, the task of object detection primarily falls to the decoder component, which is conventionally structured with six layers. Despite the uniformity in task objectives and training approaches across these layers, the decoder is pivotal for three key functions: (1) leveraging cross−attention to foster comprehensive interactions between queries and the feature map, enabling the queries to identify object presence; (2) implementing self−attention to foster communications among queries, which enhances their ability to recognize mutual information such as object co−occurrence or redundancy in detections; (3) employing a Multi−layer Perceptron (MLP) [37] to transform queries into precise object classifications and their respective bounding boxes. Each layer of the decoder refines its predecessor’s output in a successive, residual fashion, thereby incrementally enhancing the final output. Observation of the output progression across the decoder layers reveals that certain objects identified in the initial layers may be absent in the output of the sixth layer.

When analyzing the outputs from each layer of the decoder, an interesting observation emerges: objects that are identified in the early stages may vanish by the time we reach the sixth layer. This suggests that the ability to retain the detection of certain objects diminishes across the layers. The ability of subsequent layers to capture intricate details or smaller objects during feature extraction may weaken, resulting in decreased efficiency in object detection.

In this study, we revisit the decoder’s architecture by introducing an auxiliary decoder model dubbed Recursive Small Object Prediction (RSP). The RSP model amalgamates the predictions from each of the decoder’s six layers, culminating in a unified predictive output for the model. The two-stage filtering method enables RSP to effectively harmonize the outputs from these six layers, as illustrated in Figure 3.

In the first stage of the process, we sorted nine hundred predictions by their confidence scores, reviewing the predicted outcomes layer by layer to select the top−k predictions—specifically, the top three hundred—based on their probabilities. This step yielded 1800 bounding boxes in total. Then, in the second stage, aimed at recognizing small objects, the bounding boxes obtained from the initial stage were regressed and aligned back to the input image. We then measured the size of each bounding box, comparing it to the total area of the image. A particular focus was placed on bounding boxes whose dimensions were less than 32 *×* 32 pixels, assessing if the square root of the ratio of their area relative to the overall image area fell below 0.03. Bounding boxes that satisfied this requirement were retained for further analysis.

The approach includes an additional refinement to the backbone network. For the bounding boxes that passed second−stage filtering, the model captures positional data related to each box to align them with the four feature layers of the backbone network, leveraging these specific tokens as inputs for the encoder. This creates a recursive operation aiming to amplify the features of small objects, transforming the decoder’s output back into the encoder’s input. Through implementing multi−stage prediction filtering and integrating the filtered outcomes with the backbone network, the model is empowered with enhanced feature representations. This, in turn, bolsters the model’s precision in detecting small objects and improves its overall capability in managing tasks related to small object detection.

To preserve the integrity of high−level semantic information and bolster the expressiveness of low−level feature information, the study opts to extract tokens from the latter. This technique guarantees that the high−level semantic content remains undiluted while simultaneously strengthening the low−level features’ spatial information representation. This strategy markedly diminishes the computational demand of the RSP. The adopted method thereby facilitates the model’s enhanced comprehension and detection of complex environments and small subject matters, all with impeccable sentence structure and grammatical precision.

### 3.3. Normalized Wasserstein Distance

The loss function composition in DETR is established through a linear amalgamation of classification loss and coordinate loss [38,39,40,41], with the coordinate regression segment utilizing GIOU loss. Notably, this configuration does not incorporate specific optimizations for small objects. *IOU* loss [42], widely adopted as a positional loss function in object detection assignments, quantifies the divergence between the predicted bounding boxes and the actual ones. The computation for *IOU* is executed according to the subsequent formula:(2)$$IOU=\frac{A\cap B}{A\cup B}$$

The conventional Intersection over Union (*IOU*) and its derivative measures are highly scale−sensitive, responding differently to objects of various sizes. Particularly, when objects of differing scales experience the same level of positional shift, the smaller objects are disproportionately affected in the resulting *IOU* calculation—as evidenced in Figure 4. For instance, a minor positional shift in a small object measuring just 6 × 6 pixels leads to a drastic reduction in the *IOU* score, which decreases from 0.53 to 0.14. In contrast, an object of a larger scale, such as 24 × 24 pixels, experiences merely a modest change in its *IOU* score, which dips from 0.85 to 0.62 under identical conditions of positional displacement.

To bolster the accuracy of the model when identifying small objects, Normalized Wasserstein Distance (NWD) has been integrated into the total coordinate loss function for a more precise assessment of the correlation between predicted and actual bounding boxes. Diverging from the Intersection over Union (*IOU*), which calculates the Jaccard index—a measure of similarity for two finite sample sets—Normalized Wasserstein Distance (NWD) offers an advanced metric that originates from the concept of Wasserstein distance. Unlike the *IOU*, NWD can more accurately gauge the disparity between distributions, particularly in instances where there is no intersection. As such, NWD provides superior performance in gauging similarities among smaller objects. Drawing on Wasserstein distance’s foundation as a point−to−point metric, we can depict a rectangular bounding box R = (*c_x_*, *c_y_*, *w*, *h*) as a bidimensional Gaussian distribution.

In the description provided, (*c_x_*, *c_y_*) denotes the center of the bounding box, with *w* and *h* representing the width and height, respectively:(3)$$\mu =\left[\begin{array}{c} c_{x}\\ c_{y} \end{array}\right],\sum =\left[\begin{array}{cc} \frac{\omega^{2}}{4} & 0\\ 0 & \frac{h^{2}}{4} \end{array}\right]$$

Similarity between bounding boxes A and B can be transformed into distributional distance between two Gaussian distributions. If using the Wasserstein distance from optimal transport theory to calculate the dispatch distance, for two two−dimensional Gaussian distributions *µ*_1_ = *N*(*m*_1_, ∑1) and *µ*_2_ = *N*(*m*_2_, ∑2), the second−order Wasserstein distance between *µ*_1_ and *µ*_2_ is defined as
(4)W22(μ1,μ2)=‖m1−m2‖22+‖∑112−∑112‖F2
where *F* denotes the Frobenius norm. Furthermore, for Gaussian distributions *N_a_* and *N_b_* modeled according to bounding boxes *A* = (*cx_a_, cy_a_, w_a_, h_a_*) and *B* = (*cx_b_, cy_b_, w_b_, h_b_*), the formula in Equation (4) can be further simplified as follows:(5)$$W_{2}^{2}\left(N_{a},N_{b}\right)={\left\|\left({\left[cx_{a},cy_{a},\frac{\omega_{a}}{2},\frac{h_{a}}{2}\right]}^{T},{\left[cx_{b},cy_{b},\frac{\omega_{a}}{2},\frac{h_{a}}{2}\right]}^{T}\right)\right\|}_{2}^{2}$$

Given that the aforementioned formula represents a distance metric and cannot be used as a similarity measure. Therefore, we normalize it in its exponential form to obtain a new metric called Normalized Wasserstein Distance (*NWD*):(6)$$NWD\left(N_{a},N_{b}\right)=\exp\left(-\frac{\sqrt{W_{2}^{2}\left(N_{a},N_{b}\right)}}{C}\right)$$
where *C* is a constant closely related to the dataset. In the subsequent experiments, this paper empirically sets *C* to the average absolute size of the Visdrone [43] dataset, achieving optimal performance.

## 4. Experiments

This section presents extensive experimental results across a myriad of small target detection scenarios, showcasing the effectiveness of RSP−DETR. Ablation studies further vindicate that our architectural enhancements empower DINO to execute target domain detection with increased proficiency.

### 4.1. Datasets

In the following experiments, we utilized two public datasets for small target detection, VisDrone and HRSID [44], which are detailed as follows, The VisDrone dataset is a large−scale resource for drone−based vision applications. It features an extensive collection of scene and object categories, which include ten classes: pedestrian, person, bicycle, car, van, bus, truck, tricycle, awning and motorcycle. The dataset contains a total of 8599 UAV−captured images, subdivided into 6471 for training, 548 for validation, and 1580 for testing, each with a resolution of approximately 2000 × 1500 pixels. Within the training set, there are 540 k annotated instances. The average absolute size of targets in the VisDrone dataset is 35.8 pixels. When compared with other object detection datasets, such as PASCAL VOC [45] (156.6 pixels), MS COCO [46] (99.5 pixels), and DOTA [47] (55.3 pixels), the targets in VisDrone are notably smaller, consistent with the defined criteria for small objects. The HRSID dataset is designed for ship detection, semantic segmentation, and instance segmentation tasks within high resolution SAR imagery. It comprises a total of 5604 high resolution SAR images with 16,951 ship instances. The HRSID dataset draws inspiration from the construction process of the COCO dataset and includes SAR images of varying resolutions, polarization, sea states, sea areas, and coastal ports.

### 4.2. Implementation Details

Apart from our model, the majority of other experiments utilize ResNet−50 (pretrained on ImageNet) as the backbone. We do not alter the number of layers in the encoder and decoder for the DETR series models. Our experimental code is based on the detrex codebase. We trained our detector using the AdamW [48] optimizer. During the training process, we set the base learning rate to 0.0001 and applied weight decay of 0.0001, which helps to control model regularization and prevent overfitting. All experiments with these models were conducted on an NVIDIA RTX 3090 computer. The batch size was configured to a value of two, and the object query parameter was assigned nine hundred. The computational environment was provisioned with the deep learning framework PyTorch, version 1.10.0, complemented by CUDA version 11.3 and Python 3.8 to ensure compatibility and performance optimization for the experiments conducted.

### 4.3. Module Effectiveness Analysis

To unravel the distinct contributions of each component within our model’s architecture, we carried out a detailed investigation using the COCO, Visdrone, and HRSID datasets. These investigative efforts were aimed at dissecting the individual functionalities of various modules and evaluating their collective influence on the model’s overall effectiveness.

Further, to validate the effectiveness of SAC across different models, we conducted comparative experiments on the COCO dataset, with the results presented in Table 1. We configured the training duration for 12 epochs, with the original model employing ResNet50 as the backbone trained on the COCO val2017 dataset. Given that our primary detection targets are small objects, the main evaluation metrics of interest are Average Precision (AP) and AP for small objects (APs). Compared to the original DETR and its variants, the method incorporating SAC achieved an increase of 1 point in AP and 0.4 in APs. This underscores the effectiveness of SAC applied to convolutional layers.

We delved into the critical influence of the Recursive Sparse Processing (RSP) module in small target detection, specifically examining how sparsification across across feature layers of various scales influences overall performance. Within the structural framework of the DETR network, the backbone consolidates feature layers from four distinct scales, creating an input matrix of dimensions B × (H × W) × 256 that subsequently feeds into the encoder layer. Our empirical analysis embraced various contexts: one devoid of recursive sparse processing, another with sparsification implemented across all feature layers, and a third focusing on sparsification exclusively on the terminal feature layer. This was followed by an examination of their respective performance outcomes.

Table 2 showcases a side−by−side comparison conducted on the VisDrone dataset, evaluating the impact of comprehensive sparsification against that limited to lower−dimensional feature layers. The results highlight that our approach supersedes the baseline in both instances. Notably, it is the sparsification of the lower−dimensional feature layers that delivers superior outcomes, augmenting APby 3.5 points and Average Precision for Small objects (APs) by 5.4 points. These findings lend robust support to the efficacy of our RSP strategy in enhancing the detection of petite−sized objects.

As delineated in the establishment of DETR’s loss computation, the coordinate loss is determined via the Generalized Intersection over Union (GIOU) loss metric. In our detailed research, we meticulously scrutinized the hyperparameters governing the weighted blend of the GIOU loss with Normalized Wasserstein Distance (NWD). The extensive results of the analysis are documented in Table 3. Data tend to hit optimal performance when calibrating the hyperparameters near 0.3, indicating that a greater emphasis on NWD within the coordinate loss significantly bolsters the results.

### 4.4. Ablation Study

The conducted ablation study aimed to scrutinize the enhancement in the detection of small objects by sequentially incorporating three distinct components we proposed, as depicted in Table 4. The inclusion of Switchable Atrous Convolution, Recursive Small Object Prediction, and Normalized Gaussian Wasserstein Distance led to increases in the AP scores by 0.8, 2.1, and 4.4, respectively, within the Visdrone dataset’s context. The baseline data, as documented in the table, reveal that the DINO framework is suboptimal for datasets centered around small targets; conversely, our innovative approach markedly bolsters small object detection efficiency.

Notably, the explorations into Recursive Small Object Prediction and Normalized Wasserstein Distance during the ablation studies revealed significant improvements in AP and APs. Concurrently, average precision for large objects (APL) decreased by 0.7 and 1.2, respectively. The reduction in APL post Recursive Small Object Prediction is linked to an augmented count of small object boxes post the additional extraction of small target features, indirectly diminishing the count of large object boxes and, hence, APL. On the other hand, the decline following the Normalized Wasserstein Distance implementation stems from its calculation methodology—focusing solely on distribution similarity, neglecting scale discrepancy—which leads to the exclusion of certain anchors congruent with the IOU for larger targets, thereby heightening the model’s orientation towards smaller object anchors.

We also carried out ablation studies on the HRSID dataset to ensure the generalizability of the algorithm presented in this paper. The results are shown in Table 5.

Results presented in Table 5 for the HRSID dataset reveal that the baseline DINO model secured AP and APs of 57.7 and 58.1, respectively, underscoring the prevalence of small object instances within the dataset. With the integration of various modules, each performance metric saw improvements to different extents. Echoing previous discoveries, the most substantial improvement was achieved through our uniquely designed Recursive Small Object Prediction feature, which yielded remarkable gains of 5.8 in AP and 6.5 in APs, respectively.

#### Comparison of Inference Accuracy

In this section, we perform a comparative analysis of inference accuracy among various models on the VisDrone dataset. This includes RSP−DETR, RT−DETR−X [49], DINO−DETR, as well as CNN−based models such as FCOS, RetinaNet [41], QueryDet [50], Faster−RCNN, and the YOLO series of models.

Table 6 illustrates the precision of object detection on the validation set images for different models inferred on the Visdrone dataset. RSP−DETR achieves superior performance among all end−to−end detectors with a ResNet50 backbone, attaining a 2.2% increase in accuracy (33.1% AP) compared to RT−DETR−X (31.0% AP). Against conventional single−stage detectors like YOLOv5, PP−YOLOE−SOD [51], and FCOS, RSP−DETR significantly enhances precision by 7.2%, 1.3%, and 3.7% in AP, respectively. Relative to traditional two−stage detectors such as Faster R−CNN, Cascade Net [52], and QueryDet, RSP−DETR notably improves accuracy by 11.3%, 4.4%, and 4.9% AP, accordingly.

The table indicates that while there are improvements in our model’s AP and APs, its performance in APL remains inferior to that of the model before improvement and other models. This discrepancy arises because IOU demonstrates significant sensitivity differences across object scales, with predictions for small objects easily misclassified as negative samples. In contrast, NWD exhibits insensitivity to object scale variations, focusing primarily on measuring the similarity of derived Gaussian distributions.

### 4.5. Visualization

In this section, we present the visual outcomes of RSP−DETR on the Visdrone and HRSID datasets. To substantiate the efficacy of our approach, we provide a visual comparison with Ground−truth (GT) annotations, alongside detection results from DINO for reference.

Figure 5 portrays the visual results of our RSP−DETR model as evaluated on the VisDrone dataset, where a visual assessment was conducted across diverse landscapes: open fields, complex urban designs, and a bird’s−eye perspective. The visualizations reveal that while the baseline model is adept at identifying a broad array of targets, it struggles with detecting smaller and minuscule objects and occasionally produces false positives. While our RSP−DETR model encounters similar difficulties with extremely small targets, it consistently outshines the baseline in detecting small objects.

Turning to Figure 6, the HRSID data visualization is analyzed through three distinct scenarios: targets from large to medium-sized close to the shoreline, medium-sized targets situated offshore, and small targets near the shore. From the visual representations, one can observe that the baseline model’s capacity to detect larger nearshore targets falls short of expectations, due to the HRSID dataset’s imbalanced representation of ship sizes and a lesser prevalence of large vessels. However, the detection of medium−sized offshore targets is considerably precise with both the baseline and our model, likely attributed to the less complex marine backdrop. When it comes to detecting small, near−shore targets, performance dips for both models. Furthermore, the baseline model yields several false detections, a challenge exacerbated by the intricate coastal terrains and a multitude of objects that lower target contrast and complicity in distinguishing objects from the backdrop. Notably, in port areas where ship crowding is common, the challenge of occlusions and overlapping figures arises. All these results collectively indicate that our RSP−DETR model surpasses the baseline across all evaluated scenarios.

## 5. Conclusions

This research is dedicated to enhancing the detection efficiency of small targets in the DINO model, thereby broadening the general applicability of the DETR series. RSP−DETR introduces both minor and significant enhancements to the architecture. At the finer level, Switchable Atrous Convolution (SAC) substitutes standard convolutions within the backbone, leveraging the capabilities of dilated convolutions to widen the receptive field for improved small target detection and bolstering the tailored loss function for such targets. Additionally, integrating the Normalized Wasserstein Distance (NWD) into the coordinate loss notably heightens the model’s sensitivity toward small objects. On a broader scale, the model benefits from comprehensive advancements through the implementation of a recursive feature pyramid strategy. This strategy extracts predictions from each decoder layer and identifies specific areas as Regions of Interest (ROI), similar to the function of Region Proposal Networks (RPNs), which enhances the detection of small targets. These predictions are then bipartite−matched with preliminary results to identify anchors aligned with small targets, thus enhancing detection precision for diminutive objects. Thorough experimental validations and ablation studies underscore the efficacy of the introduced modifications. However, the model’s complexity is heightened by these macro−level improvements, and its performance in capturing exceedingly small targets still leaves room for enhancement. Therefore, striking a balance between performance and model stability emerges as a paramount focus for our future research endeavors.

## Figures and Tables

**Figure 1 sensors-24-03856-f001:**
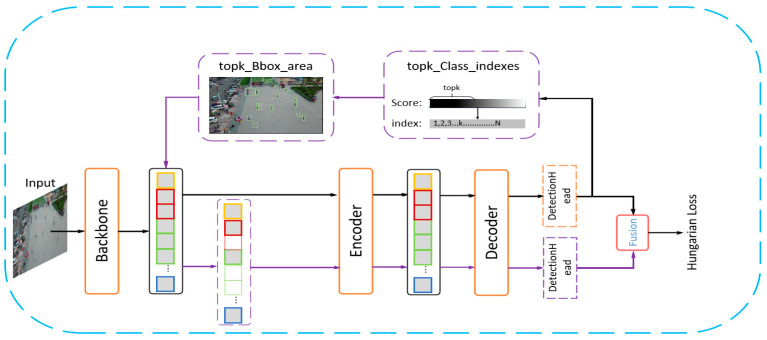
Diagram of RSP−DETR for small object detection. The backbone feature layers adopt SAC; RSP applies two−stage filtering to the decoder’s output to compensate for the lack of information on small targets. Furthermore, the loss function has been improved to enhance the performance of DINO in detecting small objects.

**Figure 2 sensors-24-03856-f002:**
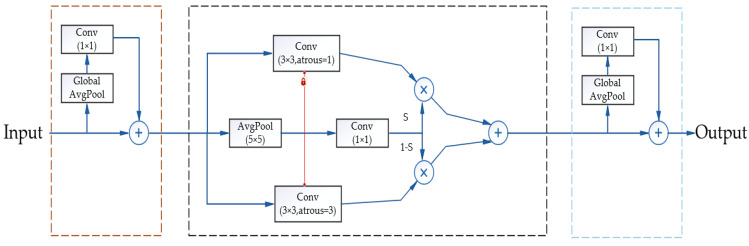
Switchable Atrous Convolution (SAC).

**Figure 3 sensors-24-03856-f003:**
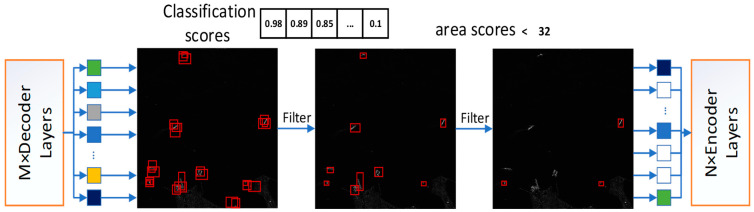
Categorical and areal information is utilized for differentiation and mapping onto the backbone.

**Figure 4 sensors-24-03856-f004:**
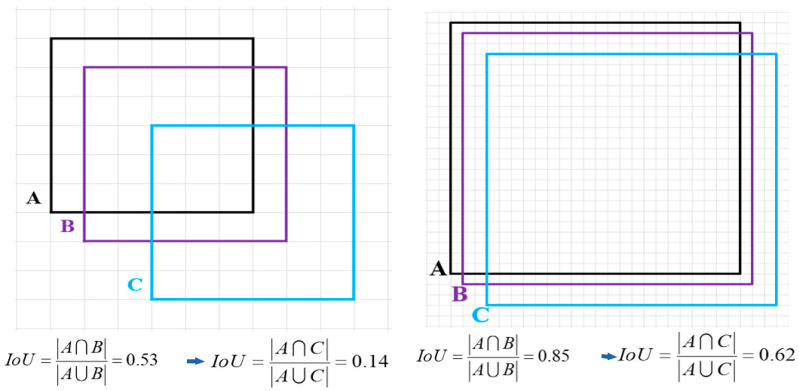
Variations in positional offset across different scales of *IOU*.

**Figure 5 sensors-24-03856-f005:**
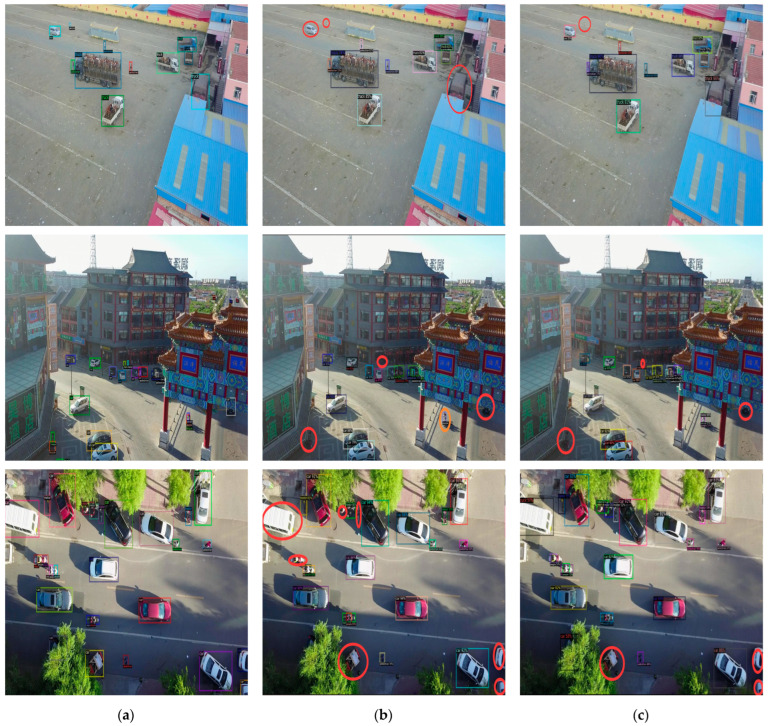
Predictive outcomes of baseline model and RSP−DETR across various scenarios: (**a**) Ground truth; (**b**) DINO; (**c**) RSP−DETR. Red circles indicate missed targets and orange circles represent erroneously detected ones.

**Figure 6 sensors-24-03856-f006:**
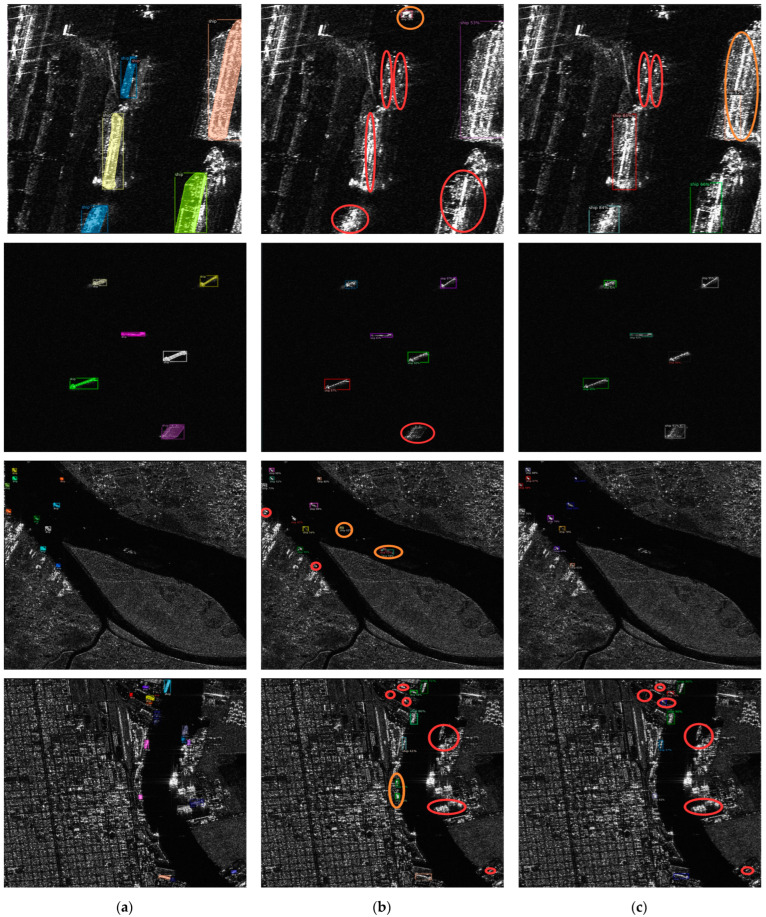
Prediction results from the baseline model and RSP−DETR on the HRSID dataset: (**a**) Ground truth; (**b**) DINO; (**c**) RSP−DETR; red circles denote missed targets, and orange circles indicate erroneously detected targets.

**Table 1 sensors-24-03856-t001:** The performance of SAC on the DINO model. The training results over 12 epochs on the COCO val2017 dataset.

Model	Epoch	AP	AP50	AP_S_	AP_M_	AP_L_
Deformable DETR [10]	12	41.0	62.6	26.4	47.1	58.0
DAB−DETR [35]	12	38.2	60.3	19.2	40.9	55.4
DN−DETR [13]	12	43.4	61.9	24.8	46.8	59.4
DINO−4scale [17]	12	49.1	66.6	32	52.3	63.2
DINO−4scale−SAC	12	50.1	67.6	32.4	53.2	65.5

**Table 2 sensors-24-03856-t002:** Comparison of sparsification by Recursive Small Object Prediction on different dimensional feature layers of the backbone.

Settings	AP	AP50	AP75	AP_S_	AP_M_	AP_L_
Baseline	27.4	50.8	25.5	17.5	42.0	51.4
complete feature set	28.6	52.3	26.9	21.0	38.9	45.4
low-level feature	30.9	53.9	30.9	22.9	41.7	52.6

**Table 3 sensors-24-03856-t003:** Analysis of the Impact of Hyperparameter λ on Model Performance.

IOU_Ratio ($\lambda_{giou}$)	AP	AP_50_	AP_75_	AP_S_	AP_M_	AP_L_
0	27.6	50.8	26.0	17.6	42.3	50.6
0.3	29.3	53.5	27.5	20.2	41.2	51.9
0.5	28.4	52.4	26.9	19.7	40.2	49.4
1	28.3	52.3	26.2	19.8	40.3	48.3

**Table 4 sensors-24-03856-t004:** Ablation experiments conducted on the Visdrone dataset.

SAC	RSP	NWD	AP	AP_S_	AP_M_	AP_L_
			27.4	17.5	**42.0**	51.4
√			28.2	18.3	41.2	**53.9**
	√		31.8	23.8	41.3	50.7
		√	29.5	20.5	41.5	50.2
√	√	√	**33.2**	**24.8**	41.8	50.8

**Table 5 sensors-24-03856-t005:** Ablation experiments conducted on the HRSID dataset.

SAC	RSP	NWD	AP	AP_S_	AP_M_	AP_L_
			57.7	58.1	60.6	31.8
√			58.3	58.6	60.8	31.9
	√		63.5	64.6	63.4	25.3
		√	59.4	59.9	63.9	24.4
√	√	√	**65.4**	**66.6**	**64.0**	**37.6**

**Table 6 sensors-24-03856-t006:** Comparison of detection accuracy among different models on the VisDrone dataset.

Model	Backbone	AP	AP50	AP75	AP_S_	AP_M_	AP_L_	FPS
FCOS [15]	ResNet–50	29.5	50.4	29.9	21.3	40.5	37.3	18
RetinaNet [41]	ResNet–50	26.2	44.9	27.1	18.6	37.5	43.8	14
QueryDet [50]	ResNet–50	28.3	48.1	28.8	20.2	37.2	46.1	16
CascadeNet [52]	ResNet–50	28.8	47.1	29.3	18.9	38.1	47.6	22
FRCNN+FPN [4]	ResNet–50	21.9	37.6	22.4	13.9	32.9	47.5	21
YOLOv5	CSP–Darknet53	26.0	42.7	27.1	15.6	42.1	52.2	43
PP–YOLOE–l [51]	CSPRepResNet	29.2	47.3	30.1	18.4	44.0	**63.3**	94
PP–YOLOE+_SOD–l	CSPRepResNet	31.9	52.1	32.6	21.7	**45.1**	60.8	90
RT–DETR–X [49]	ResNet–50	31.0	52.0	30.9	21.2	42.9	61.8	108
OURS	ResNet–50	**33.2**	**56.4**	**33.0**	**24.8**	43.8	50.8	15

## Data Availability

The data used in this study are open datasets. Data can be obtained at https://github.com/VisDrone/VisDrone-Dataset (accessed on 22 May 2024).

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
