# Peer review of "A Recursive Prediction-Based Feature Enhancement for Small Object Detection"

_sensors, 2024, doi:10.3390/s24123856_

Round 1
Reviewer 1 Report
Comments and Suggestions for Authors
This article proposes a small-scale object feature enhancement method based on the recursive prediction task to improve the detection performance of small-scale objects. Using the foundational DETR framework and improved De-Noising Anchor Boxes, seamless object detection is performed. In addition, this paper adopts a Switchable Atrous Convolution mechanism with adaptive dilated convolution to increase the receptive field and improve the inherent feature extraction ability of the primary network for small objects. The study describes the creation of a Recursive Small Object Prediction (RSP) module, which greatly improves the network's proficiency in identifying relevant features of small objects. This article is innovative and interesting. However, The following are some issues that require descriptions/clarifications.
1. The abbreviation of a specialized vocabulary should be provided in its full version the first time it appears, such as R-CNN, SSD, and YOLO, etc.
2. The logic of the Introduction is confusing and the readability is poor. In addition, the Introduction is quite long.
3. The framework diagrams of this paper, such as 1-3, are too simple. Necessary annotations in the diagram can help readers understand. On the other hand, Figure 3 does not reflect the recursive processing of the Recursive RSP.
4. In the text, subscripts are not expressed correctly. On the left side of (5), it is not N1 and N2, but Na and Nb.
5. In Table 3, the authors mentioned that “when the hyperparameter values approach 0.3, the data yields optimal results”. Why? The authors compared four groups of experiments with IOU ratios of 0, 0.3, 0.5, and 1, and concluded that the optimal value can be obtained when the parameter is 0.3. The authors can set the step size to select more IOU radios, and then provide curves instead of just tables of several sets of experimental results. The "optimal results" obtained through more experiments will be more convincing.
6. In Section 4.4.1, the authors mentioned that “This may be attributed to our model uncovering more accurate predictions during the optimization process, as the model optimization biases towards enhancing the precision of predicted outcomes rather than simply increasing the quantity of predictions.” We believe that the reasons why the proposed model is not as good as the pre-improvement model and other models in terms of APL performance should be analyzed through theoretical derivation or experimental verification, rather than pure guessing.
7. Lastly, the English writing needs to be improved. The readability of this manuscript is poor and hard to follow.
Comments on the Quality of English Language1. The abbreviation of a specialized vocabulary should be provided in its full version the first time it appears.
2. This is a lot of redundant content. Imprecise inflections and incorrect sentence patterns make the paper unreadable.
Reviewer 2 Report
Comments and Suggestions for Authors
1. The topic should be revised to remove 'RSP-DETR'.
2. What is DETR? and what is DINO in the Abstract.
3. Introduction, the first paragraph, the 2nd line, 'and ...and..', please rephrase it.
4. What is DETR[5] ?
6. Only Section 2.2 , discusses the small object detection. The authors fails to provide the state-of-art of the methods of dealing with small object detection. Besides, the authors did not mention what kind of small object detection, with what kind of features and background.
7. what is DN-DETR[13]?
8. There is no 'the' before Figure.
9. Figure 1 is too large.
10. As shown in Figure 3, it is not a sentence.
11. Fig. 3 is meaningless, please redraw.
12. Eq. (4), the paragraph below Eq.(4), there is no indention, and where should be small capital. The same as the text after Eq. (6), please check the whole manuscript.
13. Has the authors performed the real-time experiment? The field experiment must be added.
Comments on the Quality of English Language
Some English writing should improved, since it looks some places are translated from the translator.
Round 2
Reviewer 1 Report
Comments and Suggestions for Authors
The authors have addressed my concerns.
Author Response
We greatly appreciate your acknowledgment that we have addressed your concerns. Your feedback has been instrumental in refining our work, and we are grateful for the opportunity to enhance our manuscript based on your insightful suggestions. Thank you for your thorough review and constructive comments, which have undoubtedly contributed to the improvement of our paper.
Reviewer 2 Report
Comments and Suggestions for Authors The paper discusses the small object detection with recursive prediction-based feature enhancement method. Although authors make corrections on the writing and organizing of the paper. It still requires extensive work for the acceptance level.1) Abstract, 'DETR' should be explained.
2) Abstract, the first sentence, it is prejudicial with the direct DETR performance on one particular dataset. The rest does not hold the standpoint.
3) Section 1, the 2nd -4th paragraph have no clear points, what do the author try to express?
4) Section 1, the main contribution part, rephrase these points, especially point 2, how to achieve?
5) Section 2, related works, it should be combined with Section 1, since no thorough literature review is provided.
6) In the first two Sections, the logic of how many methods have been developed regarding the small object detection has not been discussed thoroughly and comprehensively.
7) The reason why choose such DETR is not clearly explained.
8) In Section 3, Figure 1 is not illustrated clearly, which is the main framework of the paper.
9) Section 3, title, Methods of what? the first sentence, '...significantly boosting the detection capabilities for small targets' does not hold.
10) 3.1, Residual Networks (ResNet) should be shown in the first place.
11) Overall, the paper should be rewritten with professional English polishing.
10( Comments on the Quality of English Language
The English of the manuscript is terrible, some of the expression is directly borrowed from the translator because certain words would not be used in the academic writing.
